# Ultrasonic Vocalizations Emission across Development in Rats: Coordination with Respiration and Impact on Brain Neural Dynamics

**DOI:** 10.3390/brainsci11050616

**Published:** 2021-05-11

**Authors:** Julie Boulanger-Bertolus, Anne-Marie Mouly

**Affiliations:** 1Department of Anesthesiology, Center for Consciousness Science, University of Michigan, Ann Arbor, MI 48109-5048, USA; 2Lyon Neuroscience Research Center, INSERM U1028, CNRS UMR 5292, University Lyon 1, 69366 Lyon, France

**Keywords:** ultrasonic vocalizations, cognitive development, respiration, brain oscillations

## Abstract

Rats communicate using ultrasonic vocalizations (USV) throughout their life when confronted with emotionally stimulating situations, either negative or positive. The context of USV emission and the psychoacoustic characteristics of the vocalizations change greatly between infancy and adulthood. Importantly, the production of USV is tightly coordinated with respiration, and respiratory rhythm is known to influence brain activity and cognitive functions. This review goes through the acoustic characteristics and mechanisms of production of USV both in infant and adult rats and emphasizes the tight relationships that exist between USV emission and respiration throughout the rat’s development. It further describes how USV emission and respiration collectively affect brain oscillatory activities. We discuss the possible association of USV emission with emotional memory processes and point out several avenues of research on USV that are currently overlooked and could fill gaps in our knowledge.

## 1. Introduction

Rats emit both audible and ultrasonic vocalizations (USV) throughout their life. USV are, by definition, emitted at a frequency higher than 19–20 kHz and are thought to be at the core of rat communication. The directionality of high frequencies facilitates the localization of the sender animal through interaural difference in the receiver while increasing their attenuation by distance, humidity, or obstacles, making them ideal for interindividual communication of small prey animals living in burrows [1]. USV are emitted when the animal is confronted with emotionally stimulating situations, but these actual situations vary between infancy and adulthood. Their psychoacoustic characteristics also change as the rat’s physiology and cognition matures. This review describes the link between USV emission, respiration, and brain dynamics throughout the rat’s development. Of note, other rodents emit USV, but their psychoacoustics, context of emission, or even link with physiology varies greatly between species [2,3], so this review is limited to rat USV.

## 2. Ethological and Anatomical Considerations for the Emission of USV

### 2.1. Vocalizing at the Different Ages of Life

Ecologically, both adult and infant rats emit USV, albeit in different contexts. However, even when emitted in response to the same artificial adverse stimulus, infant and adult USV present different characteristics [4].

#### 2.1.1. Adults

Adult rats emit two types of USV in emotionally distinct contexts. The first type of USV has a frequency of 18–32 kHz, a duration of 300–3400 ms, and shows little frequency modulation. They are referred to as 22-kHz USV and are emitted in response to antagonistic interactions with conspecifics [5,6], social isolation [7], defense against a predator [8,9], aversive stimuli [6,10,11,12], withdrawal from drugs, or after intracerebral stimulation [13,14,15,16] or mating [6,17]. As such, they are interpreted as signaling a negative emotional state or social withdrawal from an attacker or conspecific. This is further supported by the correlation between their emission rate and the intensity of the aversive stimuli [11,12], the reduction of their emission rate by systemic injection of anxiolytics [18,19], and their increase by anxiogenic drugs [19]. Within this type of USV, two subtypes can be identified: short (<300 ms) and long USV [20]. It has been suggested they could reflect distinct negative emotional states, with the short USV signaling distress without an identified source of danger and the long USV signaling distress with an identified cause [21].

A second type of adult USV has a frequency of 35–80 kHz, with or without frequency modulation, and is commonly referred to as 50-kHz USV. These USV are observed in positive social interactions, such as play [22,23,24,25] or mating [26,27,28,29,30], in anticipation of food intake or intracerebral stimulation of the reward system, or injection of addictive drugs [13,31,32,33], and are reduced by some aversive situations [13,24,34]. Therefore, they have been suggested to reflect a positive emotional state akin to human joy and laughter [35]. However, some studies also report them when rats are interacting with a complex environment [36], during short social isolation [37,38,39], during morphine withdrawal [15], or during aggressive social encounters [16,40,41,42]. In these contexts, the 50-kHz USV emission is more difficult to reconcile with a positive emotional state. Different subtypes of 50-kHz USV have been identified that can be differentially modulated by experimental interventions and could reflect different emotional states [31,43,44,45,46]. It has been suggested that non-frequency-modulated 50-kHz USV serve a social coordinating role both for social contact and food intake. On the other end, frequency-modulated USV would reflect a highly motivated state, with the inclusion of trills measuring the intensity of the positive effect [21].

#### 2.1.2. Infants

USV are also emitted by infant rats from the day after birth [47,48]. Although often called 40-kHz USV, mainly because they have often been recorded using filtering in a narrow frequency band around 40 kHz [47,49,50,51,52,53,54,55,56,57,58,59], these vocalizations span frequencies from 30 to over 100 kHz [4,60]. Ethologically, they are mostly emitted when the pups are isolated from the dam and nest [47,54] or when they are cold [49,51,56,61,62]. Additionally, they have been shown to be emitted during the extinction of operant appetitive conditioning [50], during exposure to an odor associated with gastric malaise [59], and in response to mild foot shocks [4]. The exact underlying physiological or emotional states responsible for the emission of infantile USV are still unclear. There seems to be a link with the animal’s anxiety levels, as injection of anxiolytic or anxiogenic drugs modulates USV emission [43,52,63], and some of the situations eliciting USV, such as maternal isolation, are thought to be anxiogenic. However, USV are emitted in very small amounts by pups younger than 1 week if they are isolated at a temperature equivalent to that of the nest [49,61,64], suggesting USV observed at that age in the absence of careful control of the temperature are a reaction to the lowering of the pup’s inner body temperature. Moreover, other factors affect the number of USV emitted. For example, a satiated state, the presence of the mother or littermates on the floor of the recording cage, or the presence of a threat all decrease USV production, despite having diverse consequences on the pup’s anxiety level [54,56,65,66,67]. By contrast, reisolating a pup after a short reunion with its mother potentiates the number of USV emitted, even though the mother inhibits the HPA axis of the pup (i.e., its stress response) [67,68,69,70]. Overall, while these infantile USV are usually referred to as “distress calls”, such interpretation seems to be an oversimplification and deserves further investigation.

In addition to this great variability in USV emission and our uncertainty regarding their physiological meaning and ethological purpose, the method of analysis of infantile USV varies a lot, thus increasing the challenge of comparing the results across studies. Indeed, initial studies of infantile USV have simply investigated the amount of emitted calls and reported presence of clicks—described as very short sounds heard through a bat detector—in addition to calls [47,54,56]. However, a seminal article by Brudzynski et al. in 1999 reported a great variety in the duration, frequency, bandwidth, and sonographic structure of infantile USV in rats, classifying them in 10 categories according to their shape and duration [60]. This classification has been further expanded, and subsequent articles added new categories, eventually attempting to observe the effect of various environmental factors on each call category [71,72,73,74,75]. Other studies measured the average duration, frequency, or bandwidth of the emitted USV to investigate the effect of an intervention [53,55,75,76,77].

More recently, we suggested that infantile USV might be categorized depending on their frequency rather than their shape [4]. Using mild foot shock aversive stimulation, we showed that USV could be split in two classes. The first type presents a frequency centered around 40 kHz and a duration around 200 ms, while the second type, akin to the clicks described in the earlier literature [47,56], is much shorter, being 21 ms on average and presenting a frequency of 66 kHz on average (Figure 1A). These two kinds of infant USV can either co-occur or be emitted separately. Next, we questioned if these infant USV were also emitted in a more naturalistic context when pups received rough treatment from the mother. To do so, rat pups were isolated one at a time in an unfamiliar bedding-free plexiglass cage, after which the mother was introduced. In this new environmental context, the mother spent most of her time exploring the cage, occasionally stepping on the pup, rarely nursing it, and frequently roughly transporting it. This treatment enhanced 40-kHz USV while leaving 66-kHz USV unchanged. Preliminary observations further suggested that rough handling in transport, which is a source of painful stimulation for the pup, was the most efficient stimulus for enhancing 40-kHz USV (Figure 1B).

This bimodal distribution of infant USV has been also described by others, and evidence suggests that they are differentially impacted by various interventions, such as breeding for high or low emission of 50-kHz USV in adulthood, chronic modulation of monoamines levels, or maternal presence or withdrawal [4,78,79]. Furthermore, while not explicitly described as the evidence of two categories of USV, the bimodal distribution is also visible in the data of other studies. Finally, several studies reported modulations of the average frequency and duration of USV following various interventions without reporting the distribution of the USV frequencies and durations. Therefore, in these studies, the intervention-induced modulations of the average USV frequency and duration could be the consequence of altered ratios of 40-kHz or 66-kHz USV [77,80].

Overall, the ecological relevance of different infantile USV is still unclear, regardless of the parameters analyzed. While a unification of the standards to analyze infantile USV would seem appealing, every categorization presented here has provided interesting results. However, requesting every study to analyze their USV using all these different categorizations might increase the risk of observing differences by chance (false positive increased by multiple comparisons) and likely reduce the reproducibility of the results, especially considering that USV emission is extremely variable between individuals and litters [81]. An alternative option would be for researchers to share their raw USV recordings along with their conditions of collection in a database akin to *mouseTube* to facilitate later meta-analyses [82].

### 2.2. Organs and Mechanisms Controlling USV Emission

Audible vocalizations in rats, as with human vocalizations, are the result of vibration of the vocal folds. Such audible vocalizations are usually constituted of a fundamental frequency and multiple harmonics. On the contrary, USV consist of a sound of a single frequency at a time with very little to no harmonics. The first investigation of the mechanisms of production of USV in rats dates back to the work of Laurence H. Robert in the early 1970s, showing that the frequency of USV, but not of audible vocalizations, was dramatically altered when rats were breathing light gases, suggesting a mode of production different from vibrations of the vocal folds [83]. Looking for the anatomical structure responsible for USV production, studies showed that sectioning the nerves innervating the larynx impairs the emission of USV [84,85,86], and endoscopic observations of the vocal folds during USV emission showed that they do not vibrate but are tightly opposed, leaving only a 1–2 mm opening in the back [87]. Furthermore, measuring laryngeal muscle activity suggested that they control the sound features of the emitted USV by controlling the glottal shape [88,89]. These observations led researchers to suggest that USV are produced through a hole tone, whistle-like mechanism in the larynx akin to a teakettle whistle. A hole tone whistle consists of two holes. The diameter and length of the first hole contribute to controlling the fundamental frequency of the whistle. The air flow is disturbed by passing through the first hole, and as this disturbed airflow passes through the second hole, these instabilities create vortices responsible for the sound wave [90]. For USV emission, researchers suggested that the first hole was formed at the back of the tightly opposed vocal folds, and the second hole constituted of the epiglottis and the basis of the tongue, approximately 1–1.5 mm higher in the vocal track of an adult rat [89]. This hypothesis was largely favored until recently; by studying the excised larynges of mice, Mahrt et al. showed that ultrasonic sounds comparable to USV could still be generated when the epiglottis was removed, therefore occurring without the second constriction, which is inconsistent with this hole tone whistle hypothesis [91]. The authors suggested that USV could instead be produced by an air jet from the glottis impinging on the planar wall, formed by the planar inner laryngeal wall as shown in Figure 2A (right side).

Riede et al. [92] proposed an alternative mechanism in which the exiting jet from the glottis would instead be projected onto the alar cartilage (Figure 2A, left side). In that model, sounds are generated through an edge tone whistle mechanism, with the ventral pouch formed between the vocal folds and the alar edge working as a resonator. Riede et al. [92] suggested that the impinging jet hypothesis is unlikely, as they suggest, using laryngeal airway reconstruction, that the planar surface necessary for its functioning does not exist in the rat larynx. Interestingly, a very recent study by Håkansson et al. [93] (same laboratory as [91]) confirmed the impinging jet hypothesis in the rat using in vitro larynx physiology and individual-based 3D airway reconstructions with fluid dynamics simulations. They further showed that filling the ventral pouch using aluminum spheres did not prevent the production of USV.

**Figure 2 brainsci-11-00616-f002:**
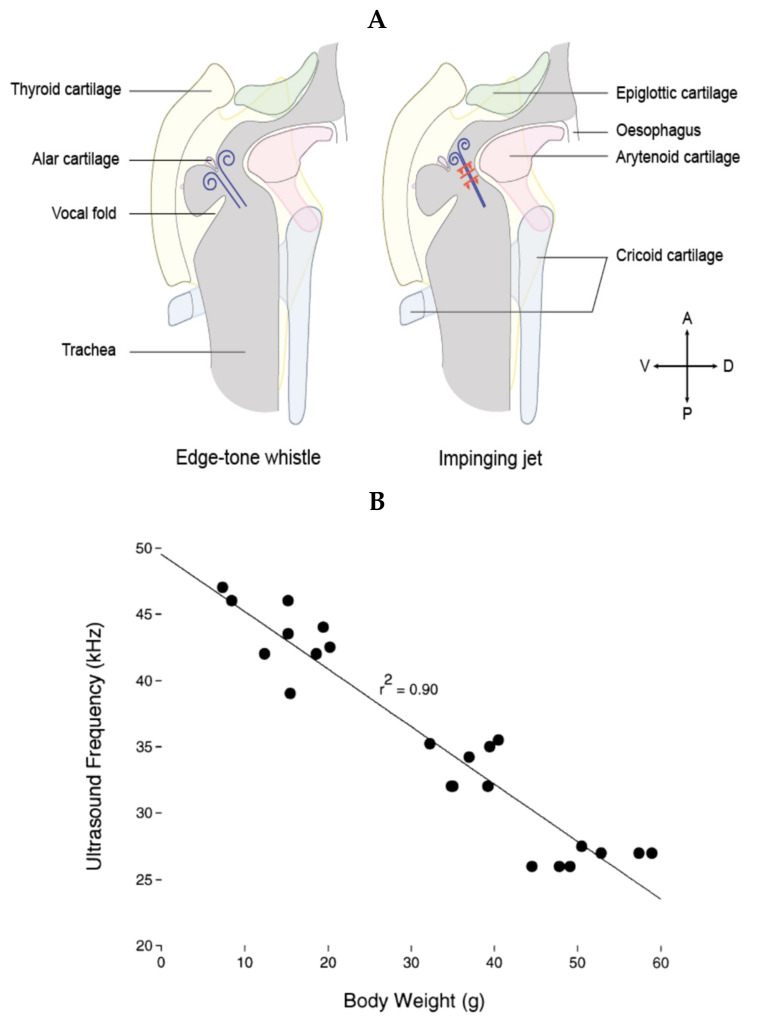
(**A**) Schematic of the proposed mechanisms for USV generation, with the edge tone whistle on the left and the impinging jet on the right. Constriction of the vocal folds leaves a small hole dorsally, through which the expiratory flow is constricted. The glottal exit jet is then either projected onto the alar edge formed by the alar cartilage (left) [92] or impinged against the thyroid wall (right) [91]. In the edge tone whistle mechanism, the ventral pouch would function as a resonator, while in the impinged jet mechanism, the upstream feedback travels back toward the glottis, generating the acoustic wave. Drawing of the larynx created from [3,94,95]. (**B**) In both models, the frequency and modulation of the USV depends on the laryngeal shape, itself determined both by the anatomy of the animal and the activity of the laryngeal muscles controlling the shape of the glottis. Accordingly, as the animal grows, the fundamental frequency of its longest USV lowers. Figure reproduced with permission from [96].

Importantly, in both models, the frequency of the emitted USV is then dependent on the distance between the glottis and the thyroid cartilage; the greater the distance, the lower the frequency of the USV. This is consistent with the observed decrease in the fundamental frequency of the rat’s longest USV as it grows (Figure 2B). Furthermore, the USV frequency would also depend on the exit speed of the jet and the diameter of the jet, which the animal can modulate using its respiration and laryngeal muscles to produce frequency-modulated USV, such as trills. Finally, it is important to note that the impinging jet mechanism produces jumps between modal frequencies (usually described as steps in USV research) that are artifactual in USV production. Therefore, the rats might not control the magnitude of the jumps, but could still control their timing by subtle changes in the expiratory flow rate or glottal configuration (Dr. Elemans, personal communication, and [93]). This is particularly important when trying to categorize 50-kHz adult USV or infant USV and highlights the need to better understand the mechanism of USV production.

### 2.3. Change in Larynx Size Changes the Sound

As the rat grows in the first weeks of its life, going from about 6 g at birth to about 150 g at 60 days old, its laryngeal shape changes, and the frequency of the USV it emits changes accordingly (Figure 2B) [96]. Furthermore, these first weeks of life witness dramatic changes in the morphology and physiology of the rat that could all alter their USV production. Rats are born deaf, with their ear canals physically closed and opening at around 14 days of life. This change affects acoustic representation in their cortex, including the representation of ultrasonic frequencies [80,97,98]. The rat also transitions from a milk-based diet to a solid diet, thus undergoing a modification of the activity of the mandibular muscles [99]. The respiration also changes as the rat grows. Its central control switches from being controlled by feedback from the pulmonary branch of the vagal nerve to a central control by the Kölliker-Fuse nuclei in the pons [100]. The respiratory motor control by the phrenic nerve also changes from a short, rapid onset burst to a long-duration discharge with a ramp [101]. This modification in the control of the respiration is associated with modifications to how the respiratory pattern is affected by the rat’s emotional state. For example, the respiratory response to an odor that has been associated with a foot shock (i.e., that became aversive) changes from a simple increase of the respiratory frequency to more modulated increases and decreases of the respiratory frequency that reflect the anticipation of a foot shock [102]. All these anatomical and physiological modifications are likely to affect the animal’s USV emission. How similar bodily changes affect the production of vocalizations has been investigated in other species, such as the marmoset [103], but a detailed analysis of changes in the mechanism of USV production is still lacking for the rat.

### 2.4. USV Emission Is Tighly Associated with the Respiratory Cycle

As mentioned earlier, USV are emitted during the expiratory phase of the respiratory cycle [12,64,104,105,106]. In the 22-kHz USV of adult rats, the length of the expiratory phase is correlated with the length of the USV and is significantly longer than the silent expiratory phases [4,12], suggesting that emitting USV lengthens the expiratory phase, and it is possible that the USV duration is limited by the physiological need to breathe [107]. Similarly, longer lower-frequency infantile USV are also correlated in length with the expiratory phase of the respiration [4]. Higher-frequency infantile USV and 50-kHz adult USV are usually much shorter than the expiratory phase in which they are emitted. However, evidence suggests that emission of 50-kHz USV still prolongs the respiratory cycle [2,4]. Importantly, the emission of USV affects the expiratory air flow. During 22-kHz USV, for example, the expiration is characterized by a drastically reduced flow rate [4,12,106]. In addition, 50-kHz USV are also emitted during a low-pressure phase following exhalation onset [2]. In the rest of this article, we gather evidence suggesting that these modifications of the respiratory output associated with USV emission are likely to influence brain function and cognitive processes [108].

## 3. Respiration Influences Brain Activity and Cognitive Functions

During nasal respiration, odorant molecules enter the nasal cavity and rhythmically stimulate olfactory receptor neurons during inhalation. This rhythmical stimulation drives oscillations time-locked to breathing cycles in the olfactory pathways, as first reported by Lord Adrian [109] and confirmed by many other studies since then [110,111,112,113,114,115]. Interestingly, olfactory receptor neurons have mechanosensitive properties and respond to changes in pressure caused by the nasal airflow [116], thus allowing entrainment of neural activity in the olfactory pathways in the absence of odor stimulation [117]. In line with these results, Fontanini and Bower suggested that “slow-wave oscillations in the cerebral cortex as a whole, including the neocortex, might be entrained and coordinated by entry of air into the nostrils” [118]. Interestingly, a number of recent papers on rodents have highlighted that, in addition to its impact on olfactory regions, nasal breathing entrains respiration-locked oscillations in several non-olfactory brain areas, such as the whisker barrel cortex [119,120], the hippocampus [121,122,123], or the prefrontal cortex [124,125,126]. In a recent study, Tort et al. [127] investigated respiration-coupled oscillations throughout the brains of freely moving mice exhibiting a broad range of respiratory frequencies and found that they could be detected in several neocortical regions, from prefrontal to visual areas and also in subcortical structures such as the thalamus, amygdala, and ventral hippocampus.

Importantly, beside entraining brain oscillatory activity at the respiratory rhythm, nasal respiration also modulates the amplitude of higher frequency oscillations. This was first demonstrated in the olfactory pathways and more specifically in the olfactory bulb, where odorant stimulation was shown to induce prominent oscillatory activity in the beta (10-35 Hz) and gamma (40-80 Hz) ranges, which alternate during a respiratory cycle [128,129,130,131]. Recent studies showed that the ability of respiratory rhythm to modulate the amplitude of fast oscillations was also observed in non-olfactory structures. Ito et al. [119] were the first to show in awake, head-fixed mice that the power of the gamma oscillations in the whisker barrel cortex was modulated in phase with breathing, a phenomenon pertaining to phase–amplitude coupling. Biskamp et al. [124] extended this finding by showing that gamma activity in the prefrontal cortex, a key associational brain region, was paced by the respiration cycle. Zhong et al. [125] further documented that respiratory rhythm modulates gamma activity in a region- and state-specific manner.

Respiration-locked oscillations and respiration-locked modulations of gamma power also occur in humans. Indeed, Herrero et al. [132], using direct intracranial recordings in humans, correlated neuronal activity with the breathing cycle and showed that the recorded signal tracked the breathing cycle across a widespread network of cortical and limbic structures. More recently, Kluger and Gross [133] used magnetoencephalography (MEG) to assess the potential influence of the respiration depth and respiration phase on the human motor system. They found coherence within the beta band to be cyclically modulated by the respiration phase. Zelano et al. [115], performing intracranial recordings from the piriform cortex, amygdala, and hippocampus in chronically implanted epileptic patients, showed that the power of oscillatory activity in different frequency bands was modulated by respiration. Importantly, this modulation was dependent on the nasal airflow, because it disappeared when patients were breathing through the mouth and no longer through the nose.

These findings suggest that the breathing rhythm, such as slow oscillatory rhythms (e.g., the theta rhythm), could help coordinate neural activity across distant brain regions by supporting the formation and synchronization of co-active cell assemblies [134,135]. In both animals and humans, oscillatory activity in the gamma frequency range has been involved in several cognitive functions, among which are attention [136,137,138], sensory perception [139,140,141], and short-term and long-term memory [138,142,143,144,145,146]. Therefore, the respiratory rhythm, through its modulation of gamma oscillatory activity, is in a good position to modulate cognitive processes.

A few recent studies have addressed this question and highlighted that some cognitive functions are modulated by the phase of the respiratory cycle. For example, Perl et al. [147] reported that participants spontaneously inhaled at non-olfactory cognitive task onset, and this resulted in improved performance accuracy in a visuospatial task. Nakamura et al. [148] used a delayed matching-to-sample visual recognition task where a test cue phase-locked to the respiratory cycle was given and showed that subjects exhibited increased reaction time and reduced accuracy when their retrieval processes encompassed the expiration to inspiration transition. Zelano et al. [115] showed that the respiratory phase has a significant influence on emotion discrimination and recognition memory. Indeed, subjects exhibited higher performances in recognizing fearful expressions and retrieving visual object memories, when target stimuli were presented during nasal inspiration rather than during expiration. In addition, the authors reported that cognitive performance significantly declined when the subjects were breathing through the mouth instead of the nose, suggesting that the route of breathing played a critical role. Similar observations were made by Arshamian et al. [149], who examined the effect of respiration on the consolidation of episodic odor memory and showed that breathing through the nose, compared with the mouth, during consolidation enhanced recognition memory, suggesting that nasal respiration is important during the critical period where memories are reactivated and strengthened.

Overall, the respiratory rhythm is transmitted from the nose to the brain through the mechanical stimulation of the olfactory neuroreceptors by the nasal airflow. In the brain, this rhythm both entrains respiration-locked oscillations and modulates higher-frequency oscillations, such as gamma oscillations, in a wide network of structures. A growing body of data suggest that through its action on brain oscillatory activity, respiration might be able to modulate some cognitive functions. Since, as described above, USV emission alters the respiratory rhythm, one can wonder if it can also modulate brain activity.

## 4. USV Emission Impacts Brain Activity

As described above, in aversive situations, such as exposure to a predator or foot shock, rats emit 22-kHz USV [69]. In the fear-conditioning paradigm, the most studied index of fear response in rats is freezing. Recent studies have shown that the expression of freezing temporally coincides with the development of sustained 4-Hz oscillations in the prefrontal–amygdala circuits, which organize the spiking activity of local neuronal populations [150,151]. Importantly, this slow oscillation is distinct from the theta rhythm and predicts the onset and offset of freezing. Interestingly, recent work has shown that freezing-related 4-Hz oscillation in the median prefrontal cortex (mPFC) is correlated with the animal’s respiratory rate, and disruption of the olfactory inputs to the mPFC significantly reduces the 4-Hz oscillation in this structure but leads to prolonged freezing periods [126]. These results indicate that the olfactory inputs can modulate rhythmic activity in the PFC and freezing behavior.

While the neural circuit involved in USV production [69,152,153,154] and the correlates of USV perception in the brain of conspecific receivers [155,156,157] are well-documented, the effect of USV production on the sender animal’s brain activity has been largely overlooked. Yet, such information is needed to better understand how the different components of fear response collectively modulate a rat’s brain neural dynamics. Importantly, as mentioned above, 22-kHz USV emission drastically slows down the animal’s respiratory rate [4,12,106], potentially disrupting the respiration-related brain rhythm described above.

In a recent study [158], we investigated the consequences of USV emission on brain oscillatory activities in the fear neural network of the vocalizing animal and assessed to what extent these changes were related to changes in the breathing rhythm (Figure 3).

To address these questions, we trained rats in an odor fear conditioning paradigm, in which an odor signaled the arrival of a foot shock a few seconds later. Fear conditioning is a widely used task in the literature of fear memories in animals, and this paradigm readily induces USV in rats. Training was conducted in an experimental cage allowing the monitoring of ultrasonic vocalizations, overt behavior, and respiration. During training, we collected local field potentials reflecting the activity of populations of neurons in the basolateral amygdala (BLA), the mPFC, and the olfactory piriform cortex (PIR). We compared the brain oscillatory activity power in different frequency bands during sequences of USV calls versus sequences without USV (silent sequences). We showed that during USV emission, the activity power in the delta (0–5 Hz), beta (15–40 Hz), and gamma (40–80 Hz) bands increased in the recorded network, while the theta (5–15 Hz) activity power decreased (Figure 4A).

We then assessed the relationship between the frequency of oscillatory activity in the delta and theta ranges and breathing frequency (which varied between 0.5 and 10 Hz), as well as the impact of USV emission on this coordination. We showed that during silent respiratory sequences, delta oscillatory activity was coupled with the breathing rhythm in the recorded network. This coupling faded away during USV **(**Figure 4B).

Finally, we assessed whether the amplitudes of the beta and gamma oscillations were modulated by the phase of the respiratory cycle (inspiration versus expiration). We showed that during silent sequences, the beta and gamma activity power was strongly modulated by the phase of the respiratory cycle. The emission of USV was associated with drastic changes in the course of time of this modulation (Figure 4B).

We proposed that the deep slow-down of the respiratory rate added to the reduction of airflow through the nose during USV calls [2,88] might be responsible for the loss of coupling between the nasal rhythm and delta oscillation. This would result in an increase in the brain’s delta oscillations power and an enhancement of the beta and gamma activity power during exhalation. We suggested that the window of a USV call, with its associated changes in nasal airflow, triggers a specific combination of brain oscillatory activities that might enhance plasticity at critical nodes of the network and ultimately strengthen long-term fear memories. Interestingly, we showed that the amount of ultrasonic vocalization during training was a good predictor of the animals’ learned fear response, measured 24 h later. The higher the number of ultrasonic vocalizations in training, the stronger the learned fear response. Hence, USV calls might result in a differential gating of information within the fear neural network, thus potentially modulating later fear memory and expression.

It is important to point out that while we described important differences in the emission of USV between infancy and adulthood, the consequences of USV emission on the infant brain remains entirely unknown at this point. Aversive events that induce USV have drastically different consequences at these two stages of development, and their memories also differentially affected the animal’s subsequent behavior [159,160]. The 22-kHz rat vocalizations present an evolutionary counterpart to human crying [161], and human infant crying and rat pup USV have been suggested to share some similarities [72]. Understanding the influence of USV, and therefore possibly cries, on the memorization of aversive events and brain processes in general could have important consequences for clinical care management of neonates.

## 5. Conclusions

Vocalization requires precise coordination of phonation, articulation, and respiration and involves a wide neural network spanning from the forebrain to the brainstem [69,162]. The present review goes through the acoustic characteristics and mechanisms of production of USV both in infant and adult rats and emphasizes the tight relationships existing between USV emission and respiration. It also provides new insights on how USV and respiratory rhythm collectively influence coordinated brain activity within the neural networks underlying defensive and emotional states. Better knowledge of the impact of ultrasonic vocalizations on brain neural dynamics is particularly relevant for rodent models of human neuropsychiatric disorders, for which socio-affective communication is severely impaired [163,164].

## Figures and Tables

**Figure 1 brainsci-11-00616-f001:**
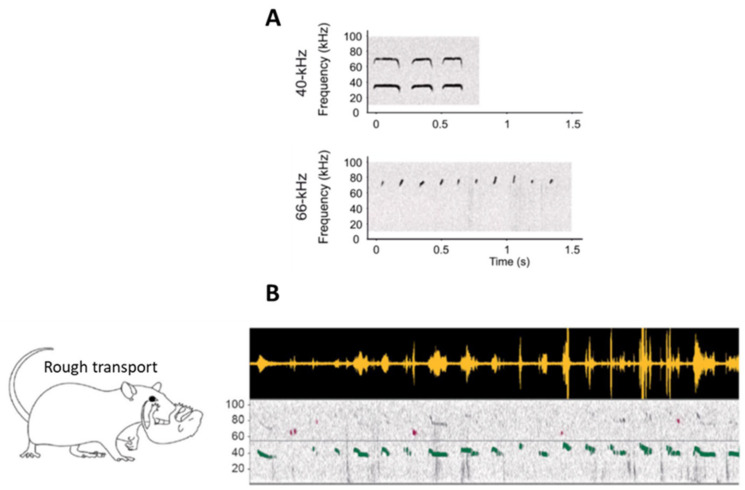
(**A**) Examples of individual sonograms of the two types of infant USV at 40 kHz and 66 kHz. (**B**) Examples of infant USV emitted in a naturalistic situation, here during rough transport by the mother. The top panel represents the raw USV signal, and the bottom panel represents the associated spectrogram (frequency in kHz as a function of time). The light horizontal gray line on the spectrogram represents the separation between 40-kHz (green) and 66-kHz USV (purple). Rough transport of the pup by the dam specifically enhanced the emission of 40-kHz USV (adapted from Boulanger-Bertolus et al. [4]).

**Figure 3 brainsci-11-00616-f003:**
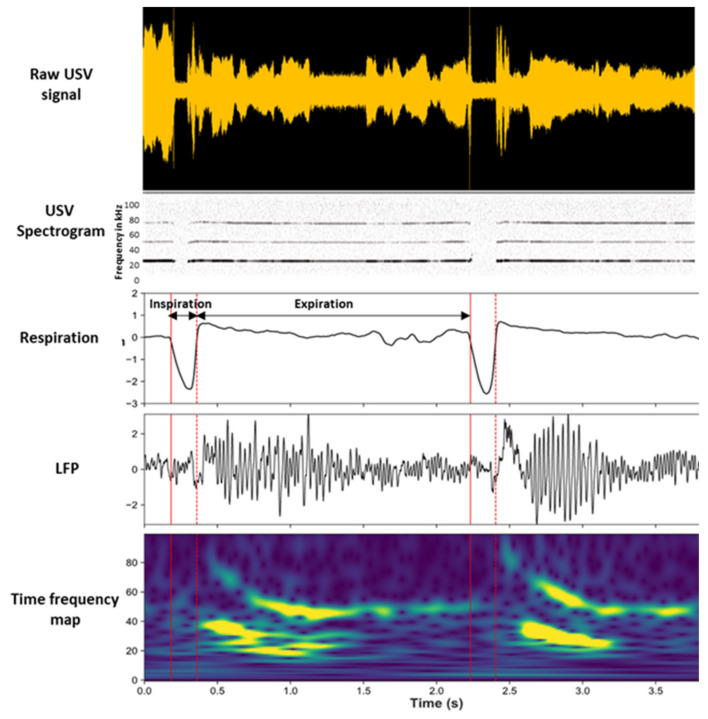
Modulation of beta and gamma oscillation power by the phase of the respiratory cycle. Individual traces represent, from the top, the raw USV calls, USV spectrogram, respiratory signal, raw local field potential (LFP) signal recorded in the piriform cortex (PIR), and its time frequency map (y-axis: LFP signal frequency in Hz; x-axis: time in milliseconds). Adapted from [158], visual abstract.

**Figure 4 brainsci-11-00616-f004:**
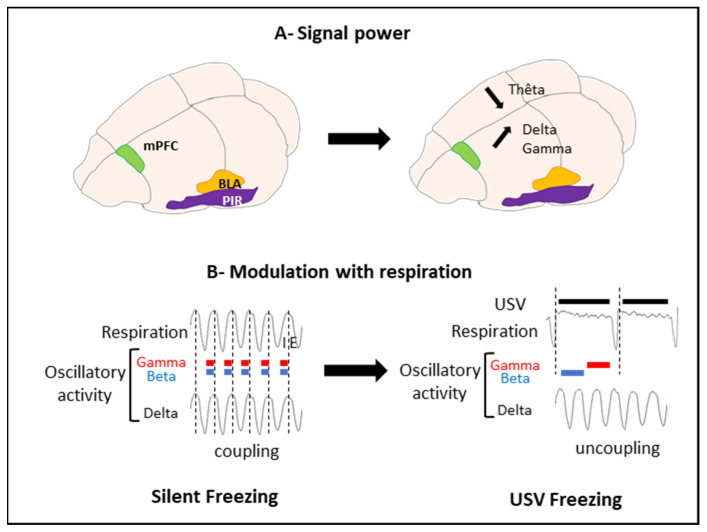
Effect of USV emission on oscillatory activity power and respiration. (**A**) Signal power. USV emission coincides with a decrease in theta power and an increase in delta and gamma power. (**B**) Modulation with respiration. Vertical dotted lines mark the transition between Inspiration (I) and Expiration (E). During silent freezing, the delta activity frequency covaries with the nasal respiratory frequency. In addition, power in the beta and gamma bands is modulated in phase with respiration, with a higher beta and gamma activity power during inspiration than expiration. During USV emission, a deep slow-down of respiratory frequency is observed, with the uncoupling between the delta frequency and respiratory frequency. Furthermore, a reorganization of beta and gamma activity power during the respiratory cycle occurs, with increased power during the first half of the expiration phase and increased gamma power during the second half of expiration (adapted from [108], Figure 10). mPFC: medial prefrontal cortex; BLA: basolateral amygdala; and PIR: piriform cortex.

## Data Availability

Not applicable.

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
