# Peer review of "Ultrasonic Vocalizations Emission across Development in Rats: Coordination with Respiration and Impact on Brain Neural Dynamics"

_brainsci, 2021, doi:10.3390/brainsci11050616_

Round 1

Reviewer 1 Report

Overall, this manuscript is very well written and the review comprehensively covers the topics presented. I have only two small requests for clarification, both of which are in figure legends:

  • In figure 1, there is a sentence (line 194) that ends abruptly and appears to be incomplete as it refers to a non-existent table 3: "In the edge-tone whistle mechanism, the ventral pouch would function as a resonator, while in the impinged jet mechanism, Table 3."
  • In figure 2 it is not obvious what the solid blue bar in the USV row represents. I assume it is the presence or absence of an ultrasonic vocalization, but given all of the discussion of frequency modulation and call categorization earlier, it would be helpful to see either the waveform or the spectrum of the actual USV.

Reviewer 2 Report

This is an interesting and well-written review on a much-needed subject. Congratulations to the authors for putting very important information together about infantile and adult vocalization and their association with respiration and brain activity. It would great if authors can discuss the sonographic structure of infantile USVs and their association with different emotional states during isolation in detail. 
